# Mutation-Specific Differences in Kv7.1 (*KCNQ1*) and Kv11.1 (*KCNH2*) Channel Dysfunction and Long QT Syndrome Phenotypes

**DOI:** 10.3390/ijms23137389

**Published:** 2022-07-02

**Authors:** Peter M. Kekenes-Huskey, Don E. Burgess, Bin Sun, Daniel C. Bartos, Ezekiel R. Rozmus, Corey L. Anderson, Craig T. January, Lee L. Eckhardt, Brian P. Delisle

**Affiliations:** 1Department of Cell and Molecular Physiology, Stritch School of Medicine, Loyola University Chicago, Maywood, IL 60153, USA; 2Department of Physiology, College of Medicine, University of Kentucky, Lexington, KY 40536, USA; deburgess@uky.edu (D.E.B.); ezekiel.rozmus@uky.edu (E.R.R.); 3Department of Pharmacology, Harbin Medical University, Harbin 150081, China; sunbinxod@gmail.com; 4Agios Pharmaceuticals, Cambridge, MA 02139, USA; danielcbartos@gmail.com; 5Cellular and Molecular Arrythmias Program, Division of Cardiovascular Medicine, Department of Medicine, University of Wisconsin-Madison, Madison, WI 53705, USA; clanders@medicine.wisc.edu (C.L.A.); ctj@medicine.wisc.edu (C.T.J.); lle@medicine.wisc.edu (L.L.E.)

**Keywords:** long QT syndrome, *KCNQ1*, *KCNH2*, K^+^ channel, heart, arrhythmia, electrocardiogram, molecular dynamics

## Abstract

The electrocardiogram (ECG) empowered clinician scientists to measure the electrical activity of the heart noninvasively to identify arrhythmias and heart disease. Shortly after the standardization of the 12-lead ECG for the diagnosis of heart disease, several families with autosomal recessive (Jervell and Lange-Nielsen Syndrome) and dominant (Romano–Ward Syndrome) forms of long QT syndrome (LQTS) were identified. An abnormally long heart rate-corrected QT-interval was established as a biomarker for the risk of sudden cardiac death. Since then, the International LQTS Registry was established; a phenotypic scoring system to identify LQTS patients was developed; the major genes that associate with typical forms of LQTS were identified; and guidelines for the successful management of patients advanced. In this review, we discuss the molecular and cellular mechanisms for LQTS associated with missense variants in *KCNQ1* (LQT1) and *KCNH2* (LQT2). We move beyond the “benign” to a “pathogenic” binary classification scheme for different *KCNQ1* and *KCNH2* missense variants and discuss gene- and mutation-specific differences in K^+^ channel dysfunction, which can predispose people to distinct clinical phenotypes (e.g., concealed, pleiotropic, severe, etc.). We conclude by discussing the emerging computational structural modeling strategies that will distinguish between dysfunctional subtypes of *KCNQ1* and *KCNH2* variants, with the goal of realizing a layered precision medicine approach focused on individuals.

## 1. Introduction

People with long QT syndrome (LQTS) are at increased risk for syncope and ventricular arrhythmias (e.g., Torsade de pointes) that may result in sudden cardiac death (SCD) (Figure 1) [1,2]. Congenital LQTS is most commonly caused by autosomal dominant mutations in the cardiac K^+^ channel genes, *KCNQ1* (LQTS type 1 or LQT1) and *KCNH2* (LQTS type 2 or LQT2) [3]. *KCNQ1* and *KCNH2* each encode a different pore-forming α-subunit of voltage-gated channel proteins, Kv7.1 and Kv11.1, respectively. The Kv7.1 and Kv11.1 channels conduct the delayed-rectifier K^+^ currents important for normal ventricular repolarization.

Most people diagnosed with LQT1 and LQT2 can be successfully managed with medications that block β-adrenergic receptors (β-blockers). Examples of β-blockers used to manage LQTS include the non-selective β-blockers nadolol and propranolol [4]. Clinical management also includes an avoidance of hypokalemia and medications that are known to prolong the QT interval [5]. People who suffered cardiac arrest or have higher-risk clinical phenotypes can be managed using implantable cardioverter defibrillators and cardiac sympathetic denervation [6,7,8]. Although people with LQT1 and LQT2 are managed similarly, they tend to show gene-specific differences in symptoms. For example, the timing of arrhythmogenic events in people with LQT1 tend to occur during the daytime, and these events are commonly triggered by activity and physical exertion [9]. In contrast, people who have LQT2 tend to suffer events in the morning, and these events associate with sudden arousal (e.g., standing, telephone ringing, alarm clock, etc.) [10]. The expressivity and penetrance of the LQT1- and LQT2-phenotypes are also influenced by the type of mutation; the location of the mutation; the impact that the mutation has on the channel function; and several additional biological, environmental, and genetic factors [1,11,12,13,14,15]. This review provides a brief background on the initial phenotypic identification of congenital LQTS and discusses the different mechanisms by which *KCNQ1* (Kv7.1) or *KCNH2* (Kv11.1) mutations can impact K^+^ currents and channel function to cause LQT1 and LQT2, respectively. Specifically, we use simplified computational models and simulations to illustrate how different LQT1- and LQT2-linked mutations can impact the human ventricular action potential (AP) duration (a cellular surrogate to the QT-interval). The AP simulations are also useful for understanding how gene-specific differences in the loss of delayed-rectifier K^+^ currents may generate distinct cellular and clinical phenotypes. We conclude with a discussion summarizing the computational approaches and strategies that are being developed to navigate the increasingly complex genotype-phenotype landscape and understand variant-specific differences in channel structure and dysfunction. The goal of this review is to compliment several existing reviews that focus on the genetics or ventricular repolarization and use of novel in vitro strategies (e.g., human inducible pluripotent stem-cell-derived cardiomyocytes) to understand inherited arrhythmia syndromes [16,17,18].

## 2. The QT-Interval: From Biomarker to Clinical Genetics

Surface electrodes placed on the skin can record the electrical activity of excitable tissues in the body including the heart. The electrocardiogram (ECG) was introduced over 120 years ago, and it was adapted to provide clinician scientists with a valuable instrument to measure the structure and function of the heart noninvasively [19,20]. Initially the ECG was used to identify cardiac arrhythmias (e.g., atrial fibrillation) [21], but it was also used to recognize ECG patterns indicative of heart disease (e.g., ischemic heart disease) [22]. In 1954, the American Heart Association published their recommendation for the standardization of the 12-lead ECG, and the 12-lead ECG remains the gold-standard for the noninvasive diagnosis of arrhythmias, conduction abnormalities, and several types of heart disease [23,24].

In 1957, Drs. Jervell and Lange-Nielsen first described a family with a “peculiar” heart disease that caused syncope and sudden death in several children [25]. The clinicians noted that ECG testing in one of the children showed an abnormally long QT-interval that worsened with exercise. Exercise seemed to trigger arrhythmias because the children suffered syncope or sudden death while running, swimming, or playing. There was no evidence that the children’s hearts were structurally or functionally abnormal, but the afflicted children were deaf. This was the first known documented case of the very rare Jervell Lange-Neilson syndrome (JLNS), the autosomal recessive form of LQTS that sometimes associates with deafness.

In the years that followed, families who had the much more common autosomal dominant form of LQTS without deafness (Romano–Ward Syndrome or RWS) were identified. In 1963 and 1964, Dr. Romano and colleagues [26], as well as Dr. Ward [27], each described families with children who suffered abnormally long QT-intervals, syncope, and SCD. In his published case report, Dr. Ward noted [27]:

“The QT-interval in the present cases is, however, so very prolonged that it must be regarded as pathological by any standard. The E.C.G. changes suggest a metabolic disorder in the myocardium which slows repolarization of the muscle after systole… Undue sensitivity of the myocardium to sympathetic stimulation is postulated. Excessive sympathetic activity may prolong QT-interval and may also contribute to ventricular fibrillation. The clinical observation that the children have improved on a drug which blocks the effect of the sympathetic at the beta receptors in the myocardium might confirm this hypothesis…”.

Clinician-scientists soon began documenting more cases of congenital LQTS and an international LQTS registry was established in 1979 [28,29]. The identification and registry of families with JLNS or RWS eventually led to an evolving diagnostic scoring scheme (the Schwartz score) that is used to identify and treat people who most likely have LQTS [7,30,31,32]. These studies also facilitated the identification of the major genes that cause JLNS and RWS [11,33,34,35,36,37]. Families with members who suffer from JLNS have loss-of-function mutations in the K^+^ channel genes that underlie the slowly activating delayed-rectifier K^+^ current (I_Ks_) in the heart (*KCNQ1* and/or *KCNE1*). Most cases of RWS are linked to loss-of-function mutations in either *KCNQ1* (LQT1) or *KCNH2* (LQT2), or gain-of-function mutations in the Na^+^ channel gene *SCN5A* (LQT3). *KCNH2* encodes the channel proteins that conduct the rapidly activating delayed-rectifier K^+^ current (I_Kr_). Phenotypical manifestations of different *SCN5A* mutations are complex and are not discussed in this review, but they have been reviewed in detail elsewhere [38,39,40].

## 3. LQT1 and LQT2 Clinical Phenotypes

Despite all the progress made over the past 50 years, accurately identifying people with LQTS is challenging [41]. The reason is mainly because clinicians use several different approaches to measure the QT-interval. Manually defining the QT-interval on an ECG is subjective and highly individualized, and computerized approaches to measure the ECG can produce widely variable results [42,43,44,45,46]. The advent of inexpensive genetic screens held the promise for improving an accurate LQTS diagnosis and patient management [47]. However, a large number of rare missense variants in LQTS-susceptibility genes exist in the general population, and as such, it is difficult to determine whether a genotype-positive test for a rare missense variant in an LQTS-susceptibility gene is causative, a genetic modifier, or neutral [48].

Additional ECG testing methods have been developed to improve the identification of people with abnormalities in ventricular repolarization, such as having the person stand from a supine position quickly [49,50]. Subsequent to standing, abnormalities in the heart rate-corrected QT-interval (QTc-interval) between the peak heart rate (QT stretching) and return to baseline (QT stunning) could be an indicator of LQTS. Another method is to measure the QTc-interval after an exercise stress test [51]. Measuring the QTc-interval immediately after an exercise stress test is particularly useful for identifying people with “concealed LQTS”. Concealed LQTS is characterized by a normal to borderline QTc-interval at rest but prolongation with provocation [52]. People with concealed LQTS tend to show abnormal QTc-intervals four minutes after stopping maximal exercise [32,51].

The exercise stress test is particularly sensitive for identifying people with LQT1 [53,54,55,56]. This is likely because the relative contribution of I_Ks_ for driving ventricular repolarization is small in basal conditions but increases during adrenergic stimulation. People with LQT2 tend to show an increase in QT-interval hysteresis with exercise [57,58]. QT-interval hysteresis is characterized by longer QT-intervals at a given RR-interval while heart rates increase during exercise, and shorter QT-intervals at the same RR-interval while heart rates decrease during recovery from exercise. QT-interval hysteresis in humans likely depends in part on the dynamic change in the relative contribution of I_Ks_ to I_Kr_ to drive ventricular repolarization before, during, and immediately after exercise.

Artificial intelligence and machine learning approaches are now being developed for the identification of people with LQTS, including concealed LQTS [59,60]. Early studies suggest that artificial-intelligence-enhanced ECGs can reliably distinguish people with concealed LQTS or the three main genotypic subgroups (LQT1, LQT2, or LQT3) with ≈80% accuracy prior to a genetic test. However, as with any emerging strategies, there are limitations to generalizing this approach, and its validation across different clinics will need to be continually assessed.

In isolation, neither standing, exercise, nor artificial enhanced ECG tests are sufficient to diagnose LQTS. However, ECG screening methods that show QTc-interval adaptation and the use of artificial intelligence will help to improve the identification of people with LQTS [32]. In the next section, we use simplified ventricular AP simulations to better illustrate how a loss in I_Ks_ or I_Kr_ can predict differences in the LQT1 and LQT2-related cellular phenotypes in basal conditions or with β-adrenergic stimulation.

## 4. Importance of I_Ks_ and I_Kr_ in Cardiac Action Potential Repolarization

The normal function of I_Ks_ and I_Kr_ is to drive cardiac repolarization and contribute to repolarization reserve. Repolarization reserve is the concept that there exists repolarization redundancy (i.e., multiple repolarizing K^+^ currents) in the normal ventricle and conduction system to aid in preventing re-entry and early-after depolarizations [61,62]. I_Ks_ and I_Kr_ can serve both primary and secondary roles in repolarization because there is a large drop in total membrane conductance that occurs during the repolarization phase of the ventricular AP.

The ventricular AP waveform is defined as having five phases: 0, 1, 2, 3, and 4. Phase 0 describes the rapid membrane depolarization; phase 1 describes a brief but rapid repolarization that creates a notch; phase 2 refers to the plateau phase; phase 3 describes the rapid repolarization phase; and phase 4 corresponds to the stable resting membrane potential (Figure 2). I_Ks_ and I_Kr_ peak at the end of phase 2 and the start of phase 3. Notice that the absolute magnitude of I_Ks_ and I_Kr_ is very small. At first glance this seems counterintuitive to their functional role to drive membrane repolarization. However, recall that the voltage response of the membrane to changes in ionic currents depends on the absolute conductance of the cell. The small I_Ks_ and I_Kr_ have a large role in driving membrane repolarization because the absolute conductance of the cell membrane drops substantially between phases 2 and 3 of the ventricular AP (Figure 2).

Computational simulations of the ventricular AP are useful for illustrating how a loss in either I_Ks_ or I_Kr_ can impact the AP duration at many different cycle lengths in basal conditions or conditions that simulate β-adrenergic stimulation. A reduction in I_Ks_ or I_Kr_ by 70% (to mimic LQT1 or LQT2, respectively) both predict a prolongation in the ventricular AP duration in basal conditions (as measured by the time to 90% repolarization or APD90) (Figure 3 and Figure 4). However, a 70% reduction in I_Ks_ predicts a smaller prolongation in the APD90 compared to a 70% reduction in I_Kr_. The importance of I_Ks_ in regulating ventricular repolarization increases following β-adrenergic stimulation [63,64,65]. β-adrenergic stimulation increases inward calcium current (I_Ca_), and the increase in inward I_Ca_ is normally offset by an increase in outward I_Ks_ to maintain or shorten the APD90 (Figure 3). Therefore, a 70% reduction in I_Ks_ predicts a longer prolongation in the APD90 during β-adrenergic stimulation compared to basal conditions. In contrast, the increase in I_Ks_ during β-adrenergic stimulation mitigates the prolongation in the APD90 in simulations of reduced I_Kr_ (Figure 4). The change in I_Ks_ with β-adrenergic stimulation helps to explain why some people with LQT1 show exaggerated QT-interval prolongation when β-adrenergic stimulation is high (e.g., during or immediately after exercise). The increase in I_Ks_ with β-adrenergic stimulation also helps to explain why some people with LQT2 have longer QT-intervals when β-adrenergic stimulation is low compared to when β-adrenergic stimulation is high (e.g., QT-interval hysteresis immediately before and after intense exercise).

## 5. Mechanistic Classification of LQT1- and LQT2-Linked Mutations

In addition to phenotypic differences in people with LQT1 and LQT2, LQT1, or LQT2, mutations can cause a loss of function in Kv7.1 and Kv11.1 by several different mechanisms. There is no one dominant disease-causing mutation for LQT1 or LQT2, and LQT1 and LQT2 each associate with hundreds of different rare mutations that span the length of the entire Kv7.1 or Kv11.1 channel protein [14,15,68,69]. It is increasingly clear that the type and location of different LQT1- or LQT2-linked mutations impact the severity of the molecular and clinical phenotype [14,70]. Kv7.1 and Kv11.1 channels are formed by the tetramerization channel proteins. Missense mutations that have dominant negative effects and interfere with the function of the wild-type (WT) protein tend to associate with worse clinical phenotypes than those that are haploinsufficient and primarily disrupt the function of the mutant protein [14,15,68,71].

To better illustrate how different types of LQT1- or LQT2-linked mutations reduce I_Ks_ or I_Kr_, it is useful to break down the biophysical components that define the amplitude of the I_Ks_ or I_Kr_ [72]. Macroscopic current (I) is equal to the number of ion channels expressed in the cell surface membrane (n), the open probability of the channels (Po), and the amplitude of ionic current through an individual channel (i), such that I = n*(Po)*(i) [73]. A mutation that reduces I_Ks_ or I_Kr_ disrupts one or more of these factors. LQT1- and LQT2-linked mutations can be categorized into four distinct classes based on the dominant mechanism that causes a loss of function. Class 1 LQT1- or LQT2-linked mutations decrease n by disrupting the synthesis of the respective Kv7.1 or Kv11.1 channel proteins; Class 2 LQT1- or LQT2-linked mutations decrease n by decreasing the insertion of Kv7.1 or Kv11.1 channel proteins in the cell surface membrane (i.e., disrupt vesicular transport/secretory pathways); Class 3 LQT1- or LQT2-linked mutations disrupt normal Po by altering normal Kv7.1 or Kv11.1 channel gating (i.e., decreasing Po); and Class 4 LQT1- or LQT2-linked mutations decrease i (i.e., disrupt ionic selectivity or K^+^ permeation through the pore) [68,70,71,74,75,76,77,78,79,80,81,82,83,84,85,86,87,88,89].

Many LQT1- and LQT2-linked mutations are radical mutations (i.e., mutations that shift the codon reading frame, disrupt mRNA transcript splicing, result in premature termination of translation, or large nucleotide deletions or insertions) and are expected to cause haploinsufficiency via a Class 1 mechanism [47,72]. Studies suggest a common mechanism by which many LQT1- and most LQT2-linked missense mutations cause a loss of channel function is to disrupt channel protein folding/trafficking/turnover and decrease the number of channel proteins expressed in the cell surface membrane (a Class 2 mechanism) [68,89]. Less common are LQT1- and LQT2-linked mutations that decrease the current by disrupting normal channel gating or single channel current levels, a Class 3 or 4 mechanism, respectively. In the next two sections we discuss how different Classes of LQT1 and LQT2 mutations associate with distinct clinical phenotypes.

## 6. LQT1-Linked Mutation Dysfunctional Phenotypes: PKA, CaM, PIP_2_, and Pleiotropy

I_Ks_ is upregulated by β-adrenergic stimulation through the activation of protein kinase A (PKA) [64]. Kv7.1 channel proteins are phosphorylated by protein kinase A (PKA) in the amino-terminus at Ser27 (S27) and S92 [90,91,92]. Phosphorylation of Kv7.1 channel proteins increases I_Ks_ by increasing the open probability of Kv7.1 channels and the amplitude of single channel currents [90]. Kv7.1 channel proteins are part of a macromolecular signaling complex that minimally includes Kv7.1, the auxiliary β-subunit, KCNE1, and the A-kinase anchoring protein, AKAP9 (Yotiao) [64,93,94]. There are also multiple interactions with other ion channel subunit proteins, but KCNE1 alters the biophysical properties of Kv7.1 channels to generate native-like I_Ks_ currents. Studies show that both KCNE1 and AKAP9 may be required for normal PKA regulation of Kv7.1 channels [64,95,96].

Genetic testing suggests 80% of suspected LQT1 mutations are missense (i.e., point mutation that leads to an amino acid substitution in frame) [47], and although in vitro studies show that different mutations reduce I_Ks_ by different absolute amounts, the reduction in I_Ks_ does not always correlate with clinical phenotypes or the risk of life-threatening events [97,98,99]. The lack of a clear association between the reduction I_Ks_ and clinical severity of LQT1 suggests additional factors are important for determining how individual LQT1 mutations impact I_Ks_ to cause life-threatening symptoms. Hiejman and colleagues (2012) demonstrated that the LQT1 missense mutation A341V-Kv7.1, which causes a severe LQT1 clinical phenotype, generates Kv7.1 channels that are insensitive to PKA. These data suggest that mutations which disrupt the PKA regulation of Kv7.1 channels can contribute to the more severe clinical phenotype in some people [100]. Barsheshet and colleagues (2012) identified several other LQT1-linked mutations that generate Kv7.1 channels which disrupt the PKA regulations of Kv7.1 channels [101]. Many of the PKA insensitive LQT1-linked mutations were found to localize to cytoplasmic loops and, similar to A341V-Kv7.1, these PKA insensitive mutations appear to confer an increased risk for life-threatening events in people.

Several *KCNQ1* mutations can also cause a “concealed LQT1” phenotype [52,66,102]. *KCNQ1* variants linked to concealed LQT1 generate Kv7.1 channels that do not cause large reductions in I_Ks_ under basal conditions, but rather, prevent the upregulation of I_Ks_ following the activation of PKA [66,102]. Studies focused on *KCNQ1* mutations that cause a concealed LQT1 phenotype used engineered phosphomimetic variants at S27 to demonstrate that these LQT1 mutations may prevent the conformational changes in Kv7.1 channels that increase I_Ks_ subsequent to PKA phosphorylation [66,102]. Unfortunately, the PKA sensitivity for only a small fraction of LQT1-linked mutations (<3%) of *KCNQ1* variants has been studied [103]. A deeper knowledge in which the mechanisms that mutant Kv7.1 channels impair Kv7.1 channel regulation by PKA is needed to better understand the potential clinical impact of different LQT1-linked mutations.

Kv7.1 channels are also modified by calmodulin (CaM). Earlier studies show that CaM is a component of Kv7.1 channels, and CaM regulates Kv7.1 channel assembly and gating [104,105]. Several Kv7.1 mutations linked to LQT1 can disrupt the CaM interaction to prevent the assembly of Kv7.1 channels. Kv7.1 channel gating is also influenced by the membrane phospholipid phosphoinositol bisphosphate (PIP_2_) [106,107]. Basic amino acid residues in the intracellular regions the Kv7.1 channels are thought to interact with PIP_2_. Several Kv7.1 missense mutations that alter these basic amino acid residues can decrease the sensitivity of Kv7.1 channels to PIP_2_ [108]. Interestingly, subsequent studies investigating protein kinase C (PKC) show PKC activation of I_Ks_ can regulate the Kv7.1 channel response to PIP_2_ [109]. After PKC activation, Kv7.1 channels are less sensitive to changes in the membrane levels of PIP_2_. Moreover, PKC partially normalizes the response of LQTS-linked Kv7.1 mutant channels to PIP_2_ by improving channel interactions with PIP_2_. Thus, the regulation of Kv7.1 channel assembly, gating, and conductance is regulated by phosphorylation, protein–protein interactions, and interactions between the channel and the phospholipids. LQTS-linked mutations in Kv7.1 channels that differentially impact the regulation of Kv7.1 channels by these mechanisms can generate distinct clinical phenotypes.

Another surprising mutation-specific phenotype associated with several LQT1-linked missense mutations is a pleiotropic effect on atrial excitability. Several LQT1-linked missense mutations can associate with familial Atrial Fibrillation (fAF) [69,110,111,112,113,114,115,116,117,118]. This is unexpected because AF is commonly attributed to reduction in the atrial refractoriness. Most of the LQT1/fAF-linked mutations generate macroscopic currents with two distinct biophysical I_Ks_ components, a voltage-dependent component, and a smaller voltage-independent component. Computational simulations of atrial APs that incorporate the functional changes of LQT1/fAF-linked mutations predict that the voltage-independent component leads to a shortening of atrial action potential duration [112,113,119]. However, computational simulations that incorporate the functional changes of LQT1/fAF-linked mutations in the ventricular action potential show they can prolong the ventricular action potential by reducing the larger voltage-dependent I_Ks_ and rendering the larger component insensitive to modulation by PKA. Although rare, LQT1/fAF-linked mutations highlight an important functional difference for I_Ks_ in the atria and ventricle, and they underscore how different *KCNQ1* mutations generate an umbrella of different LQT1-linked molecular and clinical phenotypes.

## 7. LQT2-Linked Mutation Dysfunctional Phenotypes: Trafficking in Heterotetramers

I_Kr_ is conducted by the *KCNH2*-encoded Kv11.1a and Kv11.1b channel proteins and is regulated by the *KCNE2* encoded MiRP1 auxiliary subunit [67,120,121]. Most functional expression analyses that study *KCNH2* mutations focus on studying the mutations in the Kv11.1a α-subunit [68,70]. Roughly half of LQT2-linked mutations disrupt the synthesis of Kv11.1 channel proteins and many these mutations are predicted to cause haploinsufficiency (Class 1 mechanism) [74,76,83]. The remaining LQT2-linked mutations are mostly missense, and of the >150 missense mutations in the Kv11.1a channel that have been studied, ≈90% disrupt the trafficking of Kv11.1a channels to the cell surface membrane (Class 2 mechanism) [72,75,76,77,78,79,80,81,82].

Class 2 LQT2-linked missense mutations localize to four major structural domains in the full length Kv11.1a α-subunit: the N-terminal Per-Arnt-Sim domain (PASD), the voltage-sensor domain, the pore domain, or the C-terminal cyclic nucleotide-binding domain (CNBD). Class 2 LQT2-linked mutations are postulated to decrease the folding efficiency of Kv11.1 channel proteins and increase their retention and associated degradation by cellular quality control mechanisms in the Endoplasmic Reticulum (ER). Although Class 2 LQT2-linked mutations disrupt Kv11.1 channel protein folding, studies show domain-specific differences in molecular and clinical properties [67,68,70,122,123].

LQT2 mutations in the pore domain confer the highest risk of arrhythmic events [15]. Most Class 2 mutations in the pore domain decrease the trafficking of WT-Kv11.1 channel proteins, causing dominant negative effects [68]. In contrast, Class 2 LQT2 mutations in the CNBD appear to disrupt the co-assembly of mutant LQT2 channel proteins [122,124,125]. Class 2 LQT2-linked mutations in the CNBD that disrupt Kv11.1 channel protein co-assembly are expected to be haploinsufficient [122,124,126]. Consistent with CNBD LQT2-linked mutations causing a haploinsufficiency, clinical data show some CNBD domain mutations confer a moderate to mild clinical phenotype that is unmasked in the presence of genetic modifiers [126]. Not surprisingly, immunocytochemistry studies show that cells expressing Class 2 LQT2-linked pore or CNBD mutations show dramatically different immunostaining patterns in mutant Kv11.1a channel proteins, suggesting they are regulated by distinct quality control mechanisms [125,127].

Unlike Kv7.1 channels, Kv11.1 channels do not appear to be part of a larger macromolecular complex whose proper assembly is necessary to assess function and second messenger regulation. Through this lens, functional assessment LQT2-linked mutations using the full-length Kv11.1a channel protein appears justified. However, layered on this are several channel specific aspects that confound functional pathogenic assignment using in vitro systems. In cardiomyocytes, the Kv11.1b channel protein forms tetrameric channels with Kv11.1a to conduct I_Kr_ [128,129]. Kv11.1b channel proteins are generated from an alternate start site and have a unique 5′ exon amino terminus lacking PASD [130]. Interestingly, some work has shown that Kv11.1a/Kv11.1b heterotetrametric channels have distinct biophysical current characteristics compared to Kv11.1a homotetrameric channels. In a recent study, the functional assessment of patient-specific induced pluripotent stem-cell-derived cardiomyocytes of a *KCNH2* variant located in the PASD revealed a relative increase in the transcription of Kv11.1b protein that altered I_Kr_ biophysical properties [131]. Such nuance of variant effect is not readily appreciated without the incorporation of both Kv11.1a and Kv11.1b channel isoforms, as well as their relative transcriptional regulation.

These and other features of *KCNH2* regulation add to the complexity of associating variant dysfunction with LQT2 using in vitro systems including patient-specific induced pluripotent stem-cell-derived cardiomyocytes. For example, when determining risk of arrhythmia for people with LQT2, one factor to consider is dependent on the presence or absence of sex hormones. Postpubescent males in general have lower event risk while the opposite is true for females. Moreover, sex-dependent differences may depend on mutation location, as only certain regions of the Kv11.1a channel protein appear to associate with estrogen increasing arrhythmogenic risk if the mutation stabilizes estrogen binding in the channel [132]. It has yet to be definitively determined whether these specific estrogen binding regions in Kv11.1 channel proteins are the reason for an increased risk of arrhythmic events for some LQT2 women postpartum [133]. Similar to different LQT1 mutations, the mutation-specific functional impacts that different *KCNH2* mutations have on Kv11.1 channel protein functions also generate an umbrella of different LQT2-linked phenotypes—some being more severe than others.

## 8. Visualizing Mutation-Specific Differences in Kv7.1 and Kv11.1 Channel Structures

Given the complexity and cumbersome nature of the functional assessment of individual LQT1- and LQT2-linked mutations using in vitro systems, alternative molecular-based structure–function simulations are being developed to predict the impact that individual missense variants have on Kv7.1 and Kv11.1 channel structures [134,135,136,137]. LQT1 and LQT2-linked missense mutations perturb the structure and the physiochemical properties of the channel, and these structural perturbations are expected to correlate with different mechanisms of dysfunction (e.g., Class 2, Class 3, and Class 4 mechanisms). The mutant-induced perturbations in Kv7.1 or Kv11.1 channel physicochemical properties could be orthosteric (local to the mutation position) and allosteric (far removed from the site). For mutations primarily causing orthosteric perturbations, linking structure to function changes is challenging, and mutations that cause allosteric effects are unlikely to be inferred from visualizing the Kv7.1 or Kv11.1 channel protein structures [138].

Advances in molecular simulations will be important for accurately determining orthosteric and allosteric effects that mutations have on Kv7.1 or Kv11.1 channel structure and function. Molecular simulations are widely used for predicting protein function by animating proteins based on their three-dimensional structure and molecular interactions. Recent molecular dynamic simulations of Kv7.1 and Kv11.1 channel protein structures have provided insights into how individual missense mutations perturb the channel structure in Kv7.1 and Kv11.1 [134,135,138,139,140]. As such, computational tools are beginning to serve as useful tools for mapping how mutations in Kv7.1 and Kv11.1 channel proteins may impact channel structure to cause dysfunction.

Molecular simulations rely on high (subnanometer) resolution three-dimensional structures of proteins that depict the location of individual atoms and the conformations of the amino acids. Simulations of channel structures can be used to predict changes in protein structure that underlie channel function [135], such as the shifting of the voltage-sensor in response to changes in membrane potential. Molecular dynamics and Monte Carlo simulations are among the most used for predicting conformational changes in proteins, including ion channels [141]. Molecular simulations commonly use physics-based equations and parameters to describe Å-level interactions between sets of atoms including bonds, angles, and torsions, as well as long-range potentials that encompass van der Waals forces and electrostatic interactions. Molecular simulations can predict the energies of distinct conformational states. Molecular dynamics can specifically predict how proteins dynamically ebb and flow in response to long- and short- range physical interactions by solving equations of motion. Structural simulations of LQT1- and LQT2-linked mutations are expected to identify the dynamic conformations in the Kv7.1 and Kv11.1 protein structures that associate with mutation-specific mechanisms for channel dysfunction [71,134,135,140,142,143,144].

There are presently two main challenges in using molecular dynamic simulations to accurately predict mutation-specific impacts on channel function: (1) the quality and abundance of input structural data and (2) limitations in molecular simulation. Improved resolution of protein transmembrane regions, as well as the conformations associated with gating (open, closed, inactive), will be essential for predicting channel conformations caused by missense mutations. The highest resolution structure of the transmembrane region in Kv11.1, as an example, is just under 4 Å. Average resolutions in the 3–4 Å range are generally sufficient for the correct placement of helices composing the transmembrane bundle, but will be insufficient in resolving many amino acid side chain orientations and hydrogen atoms [145]. Ambiguities in side-chain placements will almost certainly influence structure–function predictions from molecular simulations.

A key limitation to molecular simulation approaches is the mismatch between typical simulation lengths of microseconds versus the timescales of biological processes that unfold at millisecond and longer timescales [145]. Currently, bridging these disparate timescales necessitates model simplifications [146]. These approximations entail reducing the complexity of fully atomistic physics-based models by representing groups of atoms or amino acids as individual units (e.g., coarse-graining) [147], reducing the energetic detail by neglecting terms like bond vibrations (such as ‘SHAKE’), or even long-range interactions (e.g., Go models) [146,148], or by utilizing statistical and machine learning techniques in lieu of time-dependent equations of motion [149]. The balance between the aforementioned approximations and the physical detail needed to relate channel structure to phenotype has yet to be determined and will likely differ on a case-by-case basis. However, continued advances in algorithms, software, and hardware will be integral to fully realizing the potential for identifying mutation-specific differences associated with LQT1 or LQT2.

In recent years, machine learning has exploded in popularity, including its applications for predicting impacts of genetic variants on protein structures [150,151]. Applications of machine learning applied to Kv11.1 channels thus far have largely focused on predicting responses to medications that increase the risk for arrhythmias. Examples include using decision trees to determine which medications that block I_Kr_ associate with torsade de pointes or the disruption of K^+^ permeation by using neural networks [152,153]. More recent applications used ensemble methods and techniques, including naive Bayes and logistic regression, for predicting drug cardiotoxicity and variant pathogenicity [154,155,156]. Applications of machine learning to map genotype to electrophysiological cellular phenotype have been fewer in number. Along these lines, a recent study classified many different *KCNH2* variants by their potential to influence trafficking, in part by categorizing how an amino acid substitution might impact the site’s occupied volume, polarity, and steric interactions [157]. The classification of the variants included structural information that leveraged by the Kv11.1 channel structure resolved by cryo-electron microscopy [145].

While the application of machine learning approaches to voltage-gated channel function is rather recent, there is a rich history of machine learning techniques applied to protein structure predictions in general. For over a decade, machine learning techniques including Rosetta [158] and MODELLER [159] have been developed to predict protein structure from primary sequence or the three dimensional structures of homologous proteins, as has been extensively reviewed in [160]. The popularity of machine learning algorithms in the prediction of protein structure from primary sequence is reflected in the growing participation in competitions such as the Critical Assessment of Methods of Protein Structure Prediction (CASP). CASP in particular has culminated in the development of computational tools like Alpha fold [161] and RosettaTTa fold [162] for modeling protein structure [163,164], which together have been used to predict the structures of thousands of proteins they have not yet been determined by crystallography or related approaches. Additional machine learning approaches have been used for the design of proteins with user-defined tertiary structures [165]. Similarly, several approaches for predicting effects of missense mutations on protein structure have been reported [166,167,168]. Modeling mutation-specific differences in LQT1- and LQT2-linked mutations will likely benefit from recent advances in machine-learning-guided protein structure prediction.

In addition to predicting or refining protein structure, machine learning approaches have also been used to extract meaningful data from molecular dynamic simulations to predict diverse phenomena including protein–protein interactions and binding thermodynamics [169,170,171,172]. Machine learning techniques such as logistic regression, decision trees, and multilayer perceptron were recently used to predict interactions between the SARS-CoV-2 Spike Protein and Angiotensin converting enzyme 2 [173]. Additional approaches have been used to highlight physiochemical features in amyloid beta that promote their aggregation [174]. Machine learning-based refinement of computational protein structure prediction from molecular dynamic simulations has also been reported [175]. Extensions to missense variants include approaches for predicting the stability and oligomerization of epistatic enzyme [176], and impacts on protein–protein interactions [177]. Progress toward using these approaches with K^+^ channels is evident in a recent characterization of renal outer medullary K^+^ channel using in silico and experimental techniques [178]. Therefore, the wide-ranging applications of machine learning methods to molecular dynamic simulations will undoubtedly advance the forefront of understanding the structural and functional impacts of different LQT1- and LQT2-linked mutations on channel function.

## 9. Summary

Early case studies identified abnormal QT-interval prolongation as a biomarker for increased risk of SCD. Clinician-scientists were able to identify families who have autosomal recessive and dominant form of LQTS, begin the International LQTS Registry, and develop an evolving phenotypic score to identify people at risk for LQTS. The major genetic causes for LQTS were identified and most patients were found to have mutations in K^+^ channel genes important for ventricular repolarization (LQT1 and LQT2). Subsequent clinical studies identified genotype-specific differences in phenotypes and symptoms, and functional studies demonstrated that these differences are in part due to changes in I_Ks_ and I_Kr_ during β-adrenergic activation. Additional studies found subsets of LQT1- and LQT2-linked mutations showed mutation-specific differences in clinical phenotypes. Advances in molecular simulations, including molecular dynamics and machine learning, hold the promise of identifying mutation-specific differences in LQT1- and LQT2-linked mutations without the need for comprehensive in vitro testing. We expect that advances in understanding clinical phenotypes, in combination with computational approaches, will improve the clinical management of people with mutation-specific subtypes of LQT1 and LQT2.

## Figures and Tables

**Figure 1 ijms-23-07389-f001:**
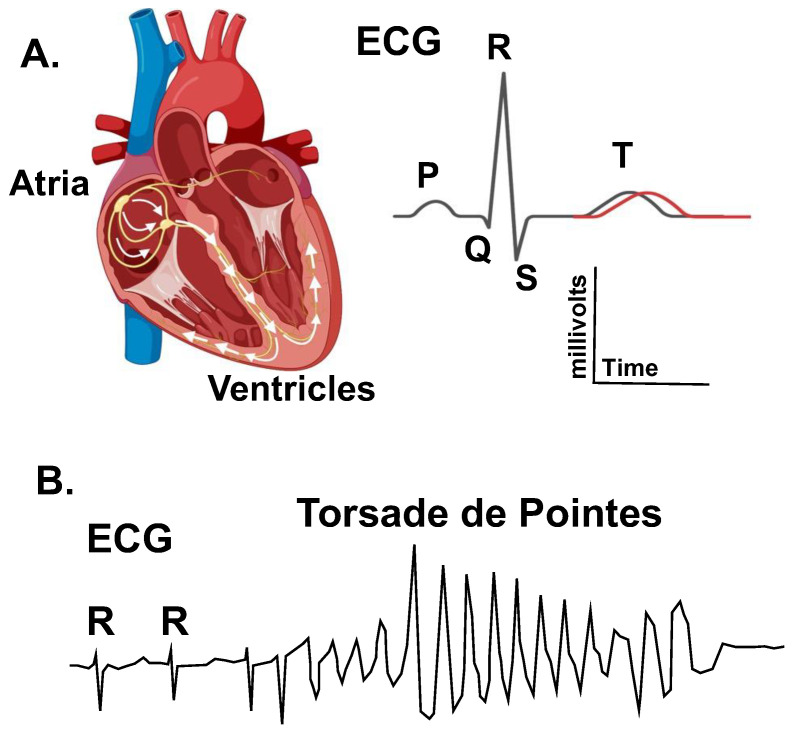
Long-QT syndrome (LQTS) is an arrhythmia disorder that sometimes causes a prolongation in the QT interval as measured on an electrocardiogram (ECG). (**A**). The left diagram shows a cross section of the heart illustrating the cardiac conduction pathway in the atrial and ventricles. The electrical activation of the heart (arrows) is measured using electrocardiography. The right image shows an ECG trace for a single cardiac cycle. The depolarization of the atria generates the P-wave, the depolarization of the ventricle generates the QRS complex, and the repolarization of the ventricle generates the T-wave. The QT-interval changes as a function of the heart rate (as measured by the RR-interval). (**A**) Prolongation in the heart rate-corrected QT-interval is a biomarker for an increased risk of polymorphic ventricular tachycardia Torsade de pointes. (**B**). Diagram of an ECG trace showing a typical Torsade de pointes arrhythmia. Torsade results in a loss in cardiac output and can cause syncope, seizures, and sudden cardiac death.

**Figure 2 ijms-23-07389-f002:**
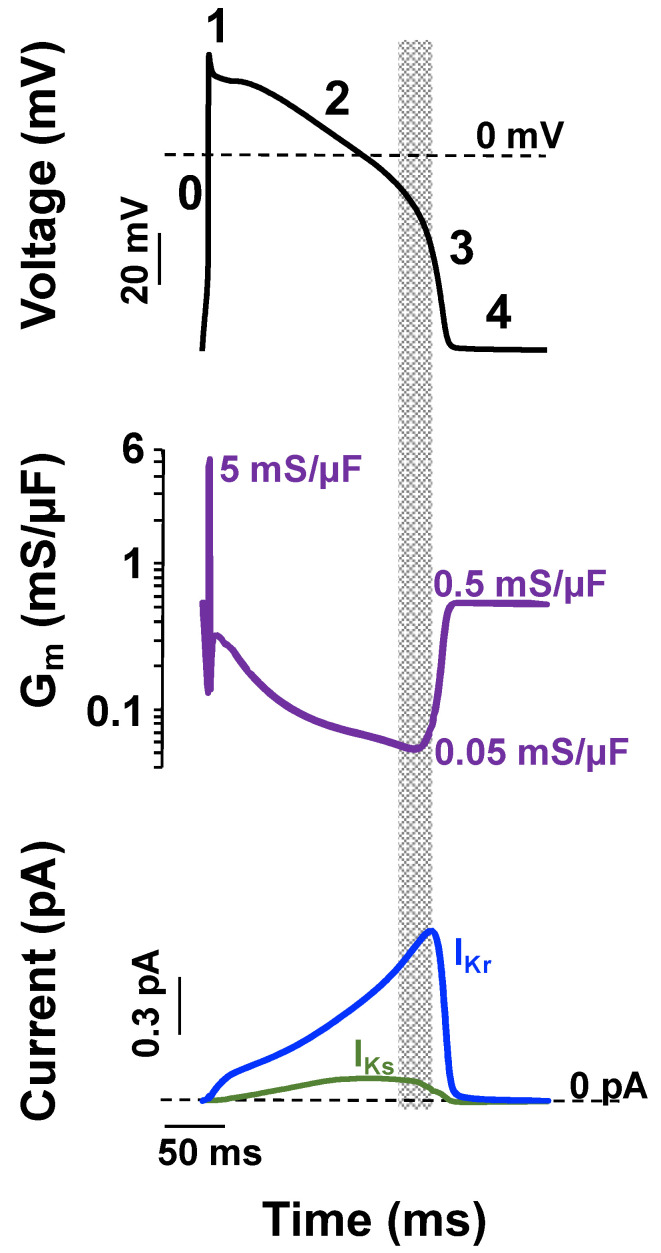
Small I_Ks_ and I_Kr_ can drive ventricular action potential repolarization and contribute to repolarization reserve because of low membrane conductance. Shown is a computational simulation for the change in membrane potential using the Soltis–Saucerman ventricular myocyte AP model [63] for normal conditions (37 °C and 5.4 mM extracellular [K^+^], black traces) driven at 1 Hz (upper panel). The phases of the AP are numbered (0–4). The middle panel shows the corresponding changes in membrane conductance (G_m_)_,_ and the bottom panel shows the corresponding I_Ks_ (green line) and I_Kr_ (blue line). The dashed lines represent 0 mV or 0 pA levels in V_m_ and macroscopic current recordings, respectively. The start of the rapid repolarization phase occurs when G_m_ is at its lowest value and I_Ks_ and I_Kr_ are near their maximal values (shaded region).

**Figure 3 ijms-23-07389-f003:**
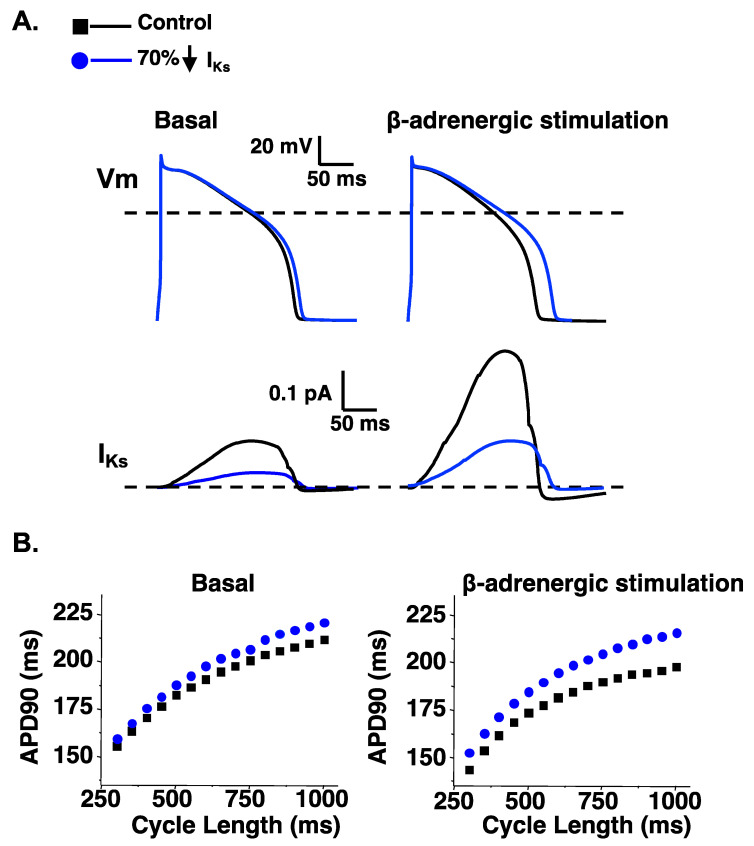
A reduction of I_Ks_ predicts a larger prolongation in the APD90 following β-adrenergic stimulation. (**A**). Representative AP waveforms and the corresponding I_Ks_ for control simulations (black line) and simulations in which the I_Ks_ component was reduced by 70% (blue line) for basal conditions (**left**) and β-adrenergic stimulation (**right**). (**B**). The steady state duration to 90% AP repolarization (APD_90_) was plotted as a function of the cycle length for basal conditions (**left**) or with β-adrenergic stimulation (**right**). Shown are the corresponding steady-state APD_90_ calculated for simulations at cycle lengths between 300 and 1000 ms for control simulations (black squares) and simulations in which the I_Ks_ component was reduced by 70% (blue circles) in basal conditions (**left**) or conditions that mimic β-adrenergic stimulation (**right**). The methodology and some of the data for producing these data is adapted from [66].

**Figure 4 ijms-23-07389-f004:**
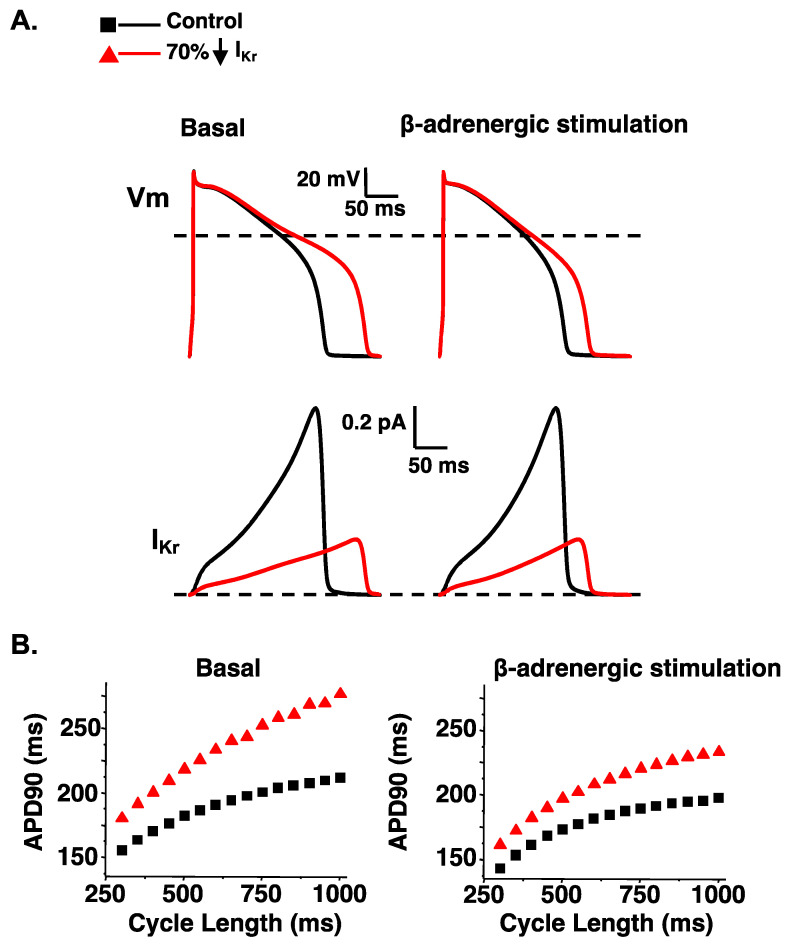
A reduction in I_Kr_ predicts a disproportionate prolongation in the AP duration in the absence of β-adrenergic stimulation. (**A**). Representative AP waveforms and the corresponding I_Kr_ for control simulations (black line); simulations in which the I_Kr_ component was reduced by 70% (red line). On the left shows basal conditions and on the right show conditions that simulate β-adrenergic stimulation. (**B**). The duration APD_90_ was plotted as a function of the cycle length for basal conditions (**left**) or with β-adrenergic stimulation (**right**). Shown are the corresponding steady-state APD_90_ calculated for simulations at cycle lengths between 300 and 1000 ms for control simulations (black squares) and simulations in which the I_Kr_ component was reduced by 70% (red triangles) in basal conditions (**left**) or conditions that mimic β-adrenergic stimulation (**right**). Some data was adapted from [67].

## Data Availability

Not applicable.

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
