# Peer review of "Mutation-Specific Differences in Kv7.1 (KCNQ1) and Kv11.1 (KCNH2) Channel Dysfunction and Long QT Syndrome Phenotypes"

_ijms, 2022, doi:10.3390/ijms23137389_

Round 1
Reviewer 1 Report
The authors focused the attention of the review in describing the role of KCNQ1 and KCNH2 IN Long QT Syndromes (LQTs), offering an historical excursus through the identification of the ECG as an important tool for the identification of abnormalities in the structure and function of the heart, and the advent of clinical genetics that led to the identification of the major genetic components responsible for the diseases. Also, an overall explanation of the electrical phenomenon during a cardiac AP are explained, however some more detailed explanations and clarifications are needed in this regard. The final part of the review offers a concise description of both KCNQ1 and KCNH2 mutations and the implication on the channel functions. Finally, in the conclusive part of the review, structural and molecular dynamics are described as examples of powerful tools to study the specific effect of mutations (LQT1 and/or LQT2-linked) on the channels structure and function. Overall, this is a timely review on an important topic.
Specific Comments
1. Pg 1. Line 38. Later in the review, the two channels encoded by the two genes are mentioned. It would best to define the two channels encoded by the gene and the current that they mediate (IKs and IKr) here and then mention again later in paragraph 1.4
2. Pg 2, Figure 1. The figure containing the two layouts, A and B, looks pretty “crowded”. Perhaps section B should be more separated from A. The ECG trace should also follow the same fashion as in A, so that ECG label is in the upper left part and then the trace. It would make it a little clearer. The cross section of the heart in Figure 1 should have the Atria and ventricles labeled or provide arrows indicating the start and propagation of the electrical activity through the heart. Otherwise, for someone who is unfamiliar with cardiac anatomy, the image will not aid much in understanding the explanation in the figure legend of the propagation of the action potential.
3. Pg 2, Line 54. One or more examples of b-blockers used as medications for patients with LQT1 and LQT2, including what class of blockers, could be reported.
4. Citation needed at line 54 “Most people diagnosed with LQT1 and…”
5. Citation needed at line 54 “Management also includes an avoidance…”
6. Citation needed at line 59 “For example, the timing of arrhythmogenic events in people…”
7. Citation needed at line 61 “In contrast, people who have LQT2 tend to suffer…”
8. Pg. 2. Incomplete or confusing sentence “Management also includes an avoidance of medications that are known to prolong the QT interval, hypokalemia, and hypomagnesemia.” If you mean avoidance of hypokalemia and hypomagnesemia, then states these first and put medications that prolong QT interval last to avoid confusion.
9. Pg. 2. Extra words “The AP simulations are also useful for understanding how gene-specific differences in in loss of delayed rectifier K+ currents may generate distinct cellular and clinical phenotypes.”
10. Pag 3, Line 102. RWS should be first reported as entire name, followed by the abbreviation in brackets - Rmano Ward Syndrome (RWS), and then used afterwards as abbreviation only afterwards (RWS)
11. Pg.4. Not clear what you mean by “In the next section, we use simplified ventricular AP simulations to illustrate how a loss in IKs or IKr can predicted differences in the LQT1 and LQT2-related cellular phenotypes.” Maybe predict instead of predicted?
12. Fig. 2. “Ohm’s law dictates that the membrane potential (Vm) is equal to the macroscopic ionic currents (I) and membrane conductance (Gm).” This is not correct. Vm is also dependent on the reversal potential of the different currents and all previous currents preceding this time point. This can easily be seen by looking at the two places where V = 0, during the upstroke the current is large and negative when V =0 and at the repolarization the current is small and positive when V = 0.
13. Pg. 5. I agree that the low overall conductance is why small K currents at the end of the ventricular action potential have such a large effect on the membrane potential. However, the presentation using a simple Ohm’s law to explain this effect is not completely accurate and more confusing than helping. Maybe instead use as an analogy the case that when one injects current in a cell the voltage response will depend on the conductance of the cell with larger voltage changes for cells with smaller conductance. This will convey the message you want without confusing the readers about how the actual membrane potential in a cell is created.
14. Pg. 5. I don’t understand this sentence “Thus, a reduction in either IKs or IKr is compensated for by the continual drop in Gm and the reserve K+ current.” Maybe add “but repolarizing the cell at a later time point, causing Long QT Syndrome”?
15. The title 1.3 “Repolarization Reserve and Ohm’s Law” doesn’t seem to correctly describe the main points of section 1.3. Section 1.3 and Figures 2, 3 and 4 mainly shows the importance of IKs and IKr currents in facilitating the repolarization phase and keeping a normal APD. For this reason, I think a title like “Importance of IKs and IKr in cardiac action potential repolarization” may be more accurate.
16. Figure 3 and 4 could be combined to create a unique figure with multiple panels so that is clearer to the reader the contribution to the two channels to the AP during the simulation and following b-adrenergic stimulation. It would be clearer when comparing the differences between the two channels explained in the paragraph.
17. Line 206. Not sure if canonical is the best word in “Simulating a reduction in IKs or IKr by 70% to mimic a canonical loss of function….”
18. Fig. 3 . Legend. Missing “of”. “A reduction in IKs predicts a disproportionate prolongation in the AP duration in the presence β-adrenergic stimulation.”
19. Pg. 6. Mix-up of alleles (DNA sequence) and channels (proteins) in “Missense mutations that have dominant negative effects and interfere with the function of the wild-type (WT) allele tend to cause worse clinical phenotypes than those that are haploinsufficient and primarily disrupt the function of the mutant allele.”
20. Pg 6, Paragraph 1.4, Line 235. Minor note to reference 65: mutations studied in this paper are relative to Kv11.1 in patients with LQT2 only. It would be appropriate to also add a reference with evidence of mutations in Kv7.1 (LQT1) that have an impact on both clinical and molecular phenotype. Lines 235-238: references should be added.
21. Line 258. Kv7.1. and Kv11.1 are introduced without any explanation.
22. Pag 8, Lines 258-264. References for each class of mutations and effect on the channels should be added.
23. Pag 9, Lines 282-283. References relative to class 3 and class 4 mutations should be added.
24. Pag 9, paragraph 1.5, line 289. cAMP should be stated with full name before using abbreviations (if list of abbreviation is not included in the review)
25. Line 308. S6 is introduced without any explanation.
26. Line 308. The clarity of this sentence could be improved “A341V-Kv7.1 lies within the S6 region, and mutations impacting neighboring amino acids although, clinically less severe than A341V-Kv7.1, tend to correlate with a more severe clinical phenotype than mutations located in other regions of Kv7.1 (e.g. all trans-membrane segments other than S6, including the extracellular and cytoplasmic loops, N-terminus, and C-terminus).”
27. Pag 12, lines 444-446. Are there any references to work previously done that show dynamic changes in the structure of the protein after mutations?
28. Pag 12, lines 447-450. Any reference to the mutations taken into the analysis for the simulations?
29. Pg.12. Not sure I understand the meaning of “Hence, structural simulations of LQT1- and LQT2-linked mutations are expected to perturb the dynamic conformations that associate with mutation-specific dysfunctional phenotypes.”.
30. Pg.12. “33 ËšA away” is not the correct way to write “33 Å away”
31. A cartoon or something to show overall channel architecture and placement of residues that are mutated would help the reader follow the discussion of mutations and structures.
32. Line 491. “biological processes that unfold at millisecond and longer rates.” Rates should be “timescales”.
33. Line 510. Maybe this sentence can be simplified? “More recent applications used ensemble methods and techniques including naive Bayes and logistic regression for predicting drug cardiotoxicity and diagnostic analysis of channelopathy genetic testing results by providing prediction of variant pathogenicity..”
34. Line 517. Not sure I understand this sentence “Uniquely, those categorizations included structural information that leveraged the intact Kv11.1 channel structure recently resolved by cryo-electron microscopy.” How could this have been done without the structure?
Author Response
Thank you for the all the helpful suggestions and important clarifications. We greatly appreciate your input on our work.
- Pg 1. Line 38. Later in the review, the two channels encoded by the two genes are mentioned. It would best to define the two channels encoded by the gene and the current that they mediate (IKs and IKr) here and then mention again later in paragraph 1.4
We have corrected this.
- Pg 2, Figure 1. The figure containing the two layouts, A and B, looks pretty “crowded”. Perhaps section B should be more separated from A. The ECG trace should also follow the same fashion as in A, so that ECG label is in the upper left part and then the trace. It would make it a little clearer. The cross section of the heart in Figure 1 should have the Atria and ventricles labeled or provide arrows indicating the start and propagation of the electrical activity through the heart. Otherwise, for someone who is unfamiliar with cardiac anatomy, the image will not aid much in understanding the explanation in the figure legend of the propagation of the action potential.
Figure 1 is reformatted to provide better spacing and labeling.
- Pg 2, Line 54. One or more examples of b-blockers used as medications for patients with LQT1 and LQT2, including what class of blockers, could be reported.
We now provide examples with new reference.
- Citation needed at line 54 “Most people diagnosed with LQT1 and…”
We have added the citation.
- Citation needed at line 54 “Management also includes an avoidance…”
We have added the citation.
- Citation needed at line 59 “For example, the timing of arrhythmogenic events in people…”
We have added the citation.
- Citation needed at line 61 “In contrast, people who have LQT2 tend to suffer…”
We have added the citation.
- 2. Incomplete or confusing sentence “Management also includes an avoidance of medications that are known to prolong the QT interval, hypokalemia, and hypomagnesemia.” If you mean avoidance of hypokalemia and hypomagnesemia, then states these first and put medications that prolong QT interval last to avoid confusion.
We corrected this sentence.
- 2. Extra words “The AP simulations are also useful for understanding how gene-specific differences in in loss of delayed rectifier K+ currents may generate distinct cellular and clinical phenotypes.”
We corrected this sentence.
- Pag 3, Line 102. RWS should be first reported as entire name, followed by the abbreviation in brackets - Romano Ward Syndrome (RWS), and then used afterwards as abbreviation only afterwards (RWS)
We corrected the abbreviation and its usage throughout.
- 4. Not clear what you mean by “In the next section, we use simplified ventricular AP simulations to illustrate how a loss in IKs or IKr can predicted differences in the LQT1 and LQT2-related cellular phenotypes.” Maybe predict instead of predicted?
We corrected this sentence.
- 2. “Ohm’s law dictates that the membrane potential (Vm) is equal to the macroscopic ionic currents (I) and membrane conductance (Gm).” This is not correct. Vm is also dependent on the reversal potential of the different currents and all previous currents preceding this time point. This can easily be seen by looking at the two places where V = 0, during the upstroke the current is large and negative when V =0 and at the repolarization the current is small and positive when V = 0. Pg. 5. I agree that the low overall conductance is why small K currents at the end of the ventricular action potential have such a large effect on the membrane potential. However, the presentation using a simple Ohm’s law to explain this effect is not completely accurate and more confusing than helping. Maybe instead use as an analogy the case that when one injects current in a cell the voltage response will depend on the conductance of the cell with larger voltage changes for cells with smaller conductance. This will convey the message you want without confusing the readers about how the actual membrane potential in a cell is created.
We agree completely and really appreciate your suggestions. This is much clearer and have incorporated this concept in the manuscript and deleted references to Ohm’s law.
- 5. I don’t understand this sentence “Thus, a reduction in either IKs or IKr is compensated for by the continual drop in Gm and the reserve K+ current.” Maybe add “but repolarizing the cell at a later time point, causing Long QT Syndrome”?
We have clarified and shortened this section.
- The title 1.3 “Repolarization Reserve and Ohm’s Law” doesn’t seem to correctly describe the main points of section 1.3. Section 1.3 and Figures 2, 3 and 4 mainly shows the importance of IKs and IKr currents in facilitating the repolarization phase and keeping a normal APD. For this reason, I think a title like “Importance of IKs and IKr in cardiac action potential repolarization” may be more accurate.
We agree and retitled the section.
- Figure 3 and 4 could be combined to create a unique figure with multiple panels so that is clearer to the reader the contribution to the two channels to the AP during the simulation and following b-adrenergic stimulation. It would be clearer when comparing the differences between the two channels explained in the paragraph.
If it is OK we prefer to keep these figures separate. Part of it is the spacing of the panels in a single figure becomes very crowded. We reworded this section so as to have the reader primarily compare the simulations for the basal condition vs. beta-adrenergic stimulation.
- Line 206. Not sure if canonical is the best word in “Simulating a reduction in IKs or IKr by 70% to mimic a canonical loss of function….”
We agree and removed this word.
- Legend. Missing “of”. “A reduction in IKs predicts a disproportionate prolongation in the AP duration in the presence β-adrenergic stimulation.”
Thank you this has been edited.
- Mix-up of alleles (DNA sequence) and channels (proteins) in “Missense mutations that have dominant negative effects and interfere with the function of the wild-type (WT) allele tend to cause worse clinical phenotypes than those that are haploinsufficient and primarily disrupt the function of the mutant allele.”
We appreciate this clarification and have modified the text accordingly.
- Pg 6, Paragraph 1.4, Line 235. Minor note to reference 65: mutations studied in this paper are relative to Kv11.1 in patients with LQT2 only. It would be appropriate to also add a reference with evidence of mutations in Kv7.1 (LQT1) that have an impact on both clinical and molecular phenotype. Lines 235-238: references should be added.
We now include the reference for LQT1.
- Line 258. Kv7.1. and Kv11.1 are introduced without any explanation.
We introduce the protein names earlier in the manuscript and tried to ease the transition between using the gene and protein nomenclature throughout.
- Pag 8, Lines 258-264. References for each class of mutations and effect on the channels should be added.
We added several different seminal publications describing the different Classes for both Kv7.1 and Kv11.1 channels.
- Pag 9, Lines 282-283. References relative to class 3 and class 4 mutations should be added.
We added references for papers studying the dysfunction of Kv7.1 channel gating and permeation.
- Pag 9, paragraph 1.5, line 289. cAMP should be stated with full name before using abbreviations (if list of abbreviation is not included in the review)
We define this abbreviation.
- Line 308. S6 is introduced without any explanation.
We define this abbreviation.
- Line 308. The clarity of this sentence could be improved “A341V-Kv7.1 lies within the S6 region, and mutations impacting neighboring amino acids although, clinically less severe than A341V-Kv7.1, tend to correlate with a more severe clinical phenotype than mutations located in other regions of Kv7.1 (e.g. all trans-membrane segments other than S6, including the extracellular and cytoplasmic loops, N-terminus, and C-terminus).”
We simplified this section for clarity.
- Pag 12, lines 444-446. Are there any references to work previously done that show dynamic changes in the structure of the protein after mutations?
We now include several references that study dynamic changes after adding mutations to channel structures.
- Pag 12, lines 447-450. Any reference to the mutations taken into the analysis for the simulations?
We removed the molecular dynamic simulation per reviewer 2’s comment.
- 12. Not sure I understand the meaning of “Hence, structural simulations of LQT1- and LQT2-linked mutations are expected to perturb the dynamic conformations that associate with mutation-specific dysfunctional phenotypes.”.
We clarified this sentence.
- 12. “33 ËšA away” is not the correct way to write “33 Å away”
We corrected the use of Angstrom.
- A cartoon or something to show overall channel architecture and placement of residues that are mutated would help the reader follow the discussion of mutations and structures.
We removed the previous Figure 5 per reviewer 2’s comment.
- Line 491. “biological processes that unfold at millisecond and longer rates.” Rates should be “timescales”.
Thank you this has been corrected.
- Line 510. Maybe this sentence can be simplified? “More recent applications used ensemble methods and techniques including naive Bayes and logistic regression for predicting drug cardiotoxicity and diagnostic analysis of channelopathy genetic testing results by providing prediction of variant pathogenicity..”
We simplified this sentence
- Line 517. Not sure I understand this sentence “Uniquely, those categorizations included structural information that leveraged the intact Kv11.1 channel structure recently resolved by cryo-electron microscopy.” How could this have been done without the structure?
We clarified this sentence.
Thank you again for a careful read of our work. It was extremely helpful in improving the clarity and quality of the manuscript. We really appreciate your input.
Reviewer 2 Report
The authors give a nice overview of the topic and the history of the LQTS, a rare disease that can lead to sudden cardiac death. They explain the cardiac action potential and the two ion channels responsible for LQT1 and LQT2 syndrome. They characterize the differences of LQT1 and LQT2 clinical phenotypes and give an overview of the different dysfunctions that mutations evoke, explaining trafficking defects as well as gating differences. In the end, they performed MD simulations and tried to explain differences in channel behavior due to mutations.
- Main point: I contacted the journal to check whether unpublished data can be published in a review article - the answer was "I am sorry to tell you thet new data that was nowhere else published before can not be published in a review article." Therefor, the MD part should be cut out of the MS.
- Could the authors please specify what they mean by “Incorporating molecular modeling to quickly predict variant-specific effects on channel function, along with deep clinical phenotyping, will improve the tailored management of people living with LQT1 or LQT2.”
- There are some small typos in there like in line 73 there are two “in” right after another, line 168 it should be “…the identification of people…”, line 496 there is twice a “range” and after “SHAKE” I’m assuming the brackets should be closed.
- It would be helpful in Figure 2 to indicate the phases 0-4 on at least the top figure.
Author Response
Thank you very much for your careful read of our work and clarifications on the use of unpublished data. We appreciate your input and suggestions.
-Main point: I contacted the journal to check whether unpublished data can be published in a review article - the answer was "I am sorry to tell you thet new data that was nowhere else published before can not be published in a review article." Therefor, the MD part should be cut out of the MS.
Response: Thank you for letting us know this. While disappointing we have revised the manuscript accordingly and removed Figure 5.
- Could the authors please specify what they mean by “Incorporating molecular modeling to quickly predict variant-specific effects on channel function, along with deep clinical phenotyping, will improve the tailored management of people living with LQT1 or LQT2.”
Response: Thank you for this comment. We clarified this sentence.
- There are some small typos in there like in line 73 there are two “in” right after another, line 168 it should be “…the identification of people…”, line 496 there is twice a “range” and after “SHAKE” I’m assuming the brackets should be closed.
Response: We have worked to carefully edit and clarify several section of the manuscript. Thank you for helping us identify these typos.
- It would be helpful in Figure 2 to indicate the phases 0-4 on at least the top figure.
Response: We now include the phases of the ventricular AP in Figure 2.
Thank you again for your input. We really appreciate your it.
Reviewer 3 Report
The review concerns the mutation-specific differences between LQT1 and LQT2 syndrome phenotypes, explaining the differences in the molecular and cellular mechanisms of the mutations.
Minor concerns:
Lines 148-149 "…is augmented during adrenergic stimulation...". The authors only mentioned the regulation of these channels by beta-adrenergic stimulation. Could be possible to mention alpha-adrenergic regulation? It so, in figure 3 a-adrenergic stimulation could be added.
In my opinion, it will be interesting if they also talk about regulation of KV7.1 by PKC and PIP2 in 1.5 section (line 286). There are some LQTS associated mutations located in the calmodulin binding site (disruption of trafficking, gating, assembly), so calmodulin could be also included in the text.
Line 348 "…IKr is conducted by the KCNH2-encoded Kv11.1a and KV11.1b channel proteins..." What about KCNE2?
Figure 5. This simulation predicts structural differences between mutations, but there is not significant differences in the distance between the mutations and V625. I would aprreciate if the authors clarify what these results mean physiologically.
Author Response
Thank you very much for the careful read of our manuscript. We now have revised per your suggestions. We really appreciate your input.
Lines 148-149 "…is augmented during adrenergic stimulation...". The authors only mentioned the regulation of these channels by beta-adrenergic stimulation. Could be possible to mention alpha-adrenergic regulation? It so, in figure 3 a-adrenergic stimulation could be added.
Response: This is an interesting point. We cannot simulate alpha-adrenergic stimulation in the simulations because we are limited to showing previously published data. Surprisingly little information about alpha adrenergic stimulation and LQT1 or LQT2 related clinical phenotypes is known.
In my opinion, it will be interesting if they also talk about regulation of KV7.1 by PKC and PIP2 in 1.5 section (line 286). There are some LQTS associated mutations located in the calmodulin binding site (disruption of trafficking, gating, assembly), so calmodulin could be also included in the text.
Response: Thank you for this important point. This was very helpful. We now include a new section discussing calmodulin, PIP2 and PKC.
Line 348 "…IKr is conducted by the KCNH2-encoded Kv11.1a and KV11.1b channel proteins..." What about KCNE2?
Response: We now include the important point that KCNE2 is part of the IKR channel complex.
Figure 5. This simulation predicts structural differences between mutations, but there is not significant differences in the distance between the mutations and V625. I would aprreciate if the authors clarify what these results mean physiologically.
Response: Per Reviewer 2’s comment, we removed the simulations from the recent manuscript draft.
Round 2
Reviewer 1 Report
Authors have responded well to comments
Reviewer 2 Report
All changes were incorporated. Fine for publication from my side.
Reviewer 3 Report
Accepted in present form